# Structural basis of adhesion GPCR GPR110 activation by stalk peptide and G-proteins coupling

Xinyan Zhu[1], Yu Qian[1], Xiaowan Li[2], Zhenmei Xu[1], Ruixue Xia[1], Na Wang[1], Jiale Liang[1], Han Yin[1], Anqi Zhang[3], Changyou Guo[3], Guangfu Wang[2] & Yuanzheng He[1] ✉

Adhesion G protein-coupled receptors (aGPCRs) are keys of many physiological events and attractive targets for various diseases. aGPCRs are also known to be capable of self-activation via an autoproteolysis process that removes the inhibitory GAIN domain on the extracellular side of receptor and releases a stalk peptide to bind and activate the transmembrane side of receptor. However, the detailed mechanism of aGPCR activation remains elusive. Here, we report the cryo-electron microscopy structures of GPR110 (ADGRF1), a member of aGPCR, in complex with $G_q$, $G_s$, $G_i$, $G_{12}$ and $G_{13}$. The structures reveal distinctive ligand engaging model and activation conformations of GPR110. The structures also unveil the rarely explored GPCR/$G_{12}$ and GPCR/$G_{13}$ engagements. A comparison of $G_q$, $G_s$, $G_i$, $G_{12}$ and $G_{13}$ engagements with GPR110 reveals details of G-protein engagement, including a dividing point at the far end of the alpha helix 5 ($\alpha$H5) of G$\alpha$ subunit that separates $G_q$/$G_s$ engagements from $G_i$/$G_{12}$/$G_{13}$ engagements. This is also where $G_q$/$G_s$ bind the receptor through both hydrophobic and polar interaction, while $G_i$/$G_{12}$/$G_{13}$ engage receptor mainly through hydrophobic interaction. We further provide physiological evidence of GPR110 activation via stalk peptide. Taken together, our study fills the missing information of GPCR/G-protein engagement and provides a framework for understanding aGPCR activation and GPR110 signaling.

Adhesion G protein-coupled receptors (aGPCRs) are membrane proteins that sense or receive information from adjacent cell-surface or extracellular matrix, and convert the stimulation into downstream signaling events mediated by heterotrimeric G-proteins[1,2]. aGPCRs play important roles in the early embryo development and are keys to many brain development events[3,4]. In addition, aGPCRs have also been linked to cancers[5]. A distinctive feature of aGPCR is the GPCR Autoproteolysis INducing (GAIN) domain right before the transmembrane domain on the extracellular side. A general activation mechanism of aGPCR

involves the removal of the GAIN domain through an autoproteolysis process which generates a short peptide called stalk peptide right before the transmembrane domain, then the stalk peptide inserts into the ligand binding pocket of the receptor and activates receptor in model called "self-activation by tethered agonist"[3,6] (Fig. 1a). The autoproteolysis allows to release the N-terminal fragment (NTF) of the receptor. The remaining N-terminal region of the receptor includes the stalk peptide. On the other hand, the C-terminal fragment (CTF) includes the 7 transmembrane helices bundle and the C-tail, and is the

[1]Laboratory of Receptor Structure and Signaling, HIT Center for Life Sciences, Harbin Institute of Technology, Harbin 150001, China. [2]Laboratory of Neuroscience, HIT Center for Life Sciences, Harbin Institute of Technology, Harbin 150001, China. [3]HIT Center for Life Sciences, School of Life Science and Technology, Harbin Institute of Technology, Harbin, China. ✉e-mail: ajian.he@hit.edu.cn

**Fig. 1 | The self-activated GPR110 is able to couple $G_q$, $G_s$, $G_i$, $G_{12}$ and $G_{13}$. a** A schematic cartoon of the self-tethering activation of GPR110. **b** Overall structures of GPR110/$G_q$, GPR110/$G_s$, GPR110/$G_i$, GPR110/$G_{12}$ and GPR110/$G_{13}$ complexes. Left side of each subpanel, orthogonal views of the cryo-EM density map of GPR110/G-protein complexes. Upper right corner of each subpanel, model of the complex in the same view and color scheme as shown in the labels.

main initiator of signal transduction. There is very limited structural information of the transmembrane domain of aGPCR except the glucocorticoid-bound GPR97[7], however, the structure of the cortisol-bound GPR97 is more like a traditional ligand-activated class A GPCR. The exact mechanism of the self-activation by tethered agonist of aGPCRs remains elusive.

GPR110, also known as ADGRF1, is an aGPCR that plays a key role in neurite growth and synaptogenesis in the cortical neurons[8]. GPR110 has been used as a model to study the self-activation by tethered agonist mechanism of aGPCRs[9]. The N-terminal truncation of GPR110 at the GPCR proteolysis site (GPS), has a robust self-activation activity in the cell-based reporter assays. Deletion of the stalk peptide leads a total abrogation of receptor activity[9]. Recently, synaptamide, an endogenous metabolite of docosahexaenoic acid (DHA), was found to be the endogenous ligand of GPR110 that binds to the GAIN domain of the receptor[8,10].

A persistent question of the GPCR field is the coupling selectivity of G proteins. There is only 4 major groups of G-proteins, $G_s$, $G_{i/o}$, $G_{q/11}$, $G_{12/13}$ whose specificities are mainly determined by the alpha subunit of $G\alpha_s$, $G\alpha_i$, $G\alpha_q$ and $G\alpha_{12/13}$ respectively[11]. On the other hand, there are more than 800 receptors in the human genome. Many receptors have the ability to couple multiple G-proteins, while a number of receptors exclusively couple to one specific G-protein. The spectrum is even more complicated for scenarios where different ligands of one receptor may induce different G-protein couplings (biased agonism). Numerous studies have shown that a simple sequence motif does not exist for receptor/G-protein recognition[12,13]. A large amount of GPCR/$G_s$, GPCR/$G_i$ and GPCR/$G_q$ complex structures are now available due to advancements in cryo-EM, however, the structural information of $G_{12/13}$ engagement are very limited.

In this work, we report cryo-EM structures of GPR110 in complex with $G_q$, $G_s$, $G_i$, $G_{12}$ and $G_{13}$ and systematically compare all major G-protein couplings in one receptor which revealed undisclosed information of G-protein coupling. The structural information provided here lays cornerstone for understanding aGPCR activation and G-protein selectivity.

## Results

### The self-activated GPR110 is able to couple with $G_q$, $G_s$, $G_i$ and $G_{12/13}$

First we asked whether the self-activated GPR110 is able to couple with four major G-proteins and activate downstream signaling pathways. We used cell-based reporter assays to examine receptor activity of different signaling pathways. The nuclear factor of activated T-cells response element (NFAT-RE) reporter assay has been established to examine $G_q$ signaling, similarly, cAMP response element (CRE), serum response element (SRE) and serum response factor response element (SRF-RE) reporter assays have been established for $G_s$, $G_i$ and $G_{12/13}$ signaling, respectively[14]. The reporter assay data show that the CTF of GPR110 has a very strong self-activation activity on the NFAT-RE, CRE, SRE and SRF-RE reporter assays, and deletion of the stalk peptide totally abrogate the activity (Supplementary Fig. 1a), indicating that the CTF of GPR110 is able to activate all 4 major G-protein signaling pathways, and the activity is mediated by the stalk peptide. Interestingly, we also observed that the full-length GPR110 has a significant amount of self-activation activity in reporter assays, particularly on the SRF-RE and SRE reporter assays, suggesting that the full-length receptor may already be activated by certain cellular factor, i.e. endogenous ligand. In fact, synaptamide, a metabolite of docosahexaenoic acid (DHA)[8,10], has been identified as a ligand of GPR110. We then asked whether the CTF of GPR110 is able to form stable complexes with $G_q$, $G_s$, $G_i$, $G_{12}$ and $G_{13}$ proteins in vitro. To increase complex stability, we used a NanoBiT tethering strategy for receptor/G-protein complex assembling. We fused the large part of NanoBiT (LgBiT)[15] to the C-terminus of the GPR110 CTF (567- 873) (Supplementary Fig. 1b) and the renovated high affinity small part of NanoBit (HiBiT) to the C-terminus of Gβ. For $G_q$ complex assembling, we used a mini-$G_q$ from the ghrelin receptor/$G_q$ complex[16]; for $G_s$ complex assembling, we used a mini-$G_s$ from the melanocortin receptor 1 (MC1R)/$G_s$ complex[17]; for $G_i$ complex assembling, we used a dominant negative version of $G_i$ which contains G203A and A326S mutations[18]. For $G_{12}$ and $G_{13}$ complex assembling, we swapped the "GGSGG" linker of mini-$G_{12}$[19] or the alpha-helical domain (AHD) of $G_{13}$ with the AHD of $G_i$, a strategy that has been

successfully used in the assembling the type 1 bradykinin receptor (B1R)/$G_q$ complex[20] (Supplementary Fig. 2). In addition, we substituted the N-terminus of $G_{12}$ and $G_{13}$ with the first 18 residues of $G_i$ to render their abilities to bind scFv16[21], a fab fragment that has been successfully used in stabilizing receptor/G-protein complex. We co-expressed receptor, Gα, Gβ and Gγ in insect Sf9 cells and purify the complex in a procedure commonly used in obtaining GPCR/G-protein complexes (see methods for detail). Nb35[22] is added in the mini-$G_s$ and mini-$G_q$ complex assembling, and scFv16 was added in the $G_i$, $G_{12}$ and $G_{13}$ complex assembling. The expression and purification data show that the CTF of GPR110 can form stable complexes with $G_q$, $G_s$, $G_i$, $G_{12}$ and $G_{13}$ (Supplementary Fig. 1c). Taken together, our data show that GPR110 is capable of coupling to all major 4 G-protein pathways, and forming complexes with $G_q$, $G_s$, $G_i$, $G_{12}$ and $G_{13}$.

## The overall architecture of GPR110/$G_q$, $G_s$, $G_i$, $G_{12}$ and $G_{13}$ complex

The complex structures of $G_q$, $G_s$, $G_i$, $G_{12}$ and $G_{13}$ are determined by the single particle analysis of cryo EM. The global resolutions for the $G_q$, $G_s$, $G_i$, $G_{12}$ and $G_{13}$ complex are 2.85 Å, 2.84 Å, 3.09 Å, 2.8 Å and 2.66 Å respectively based on the gold standard of Fourier Shell Correlation (FSC) = 0.143 criterion (Supplementary Fig. 3 to 5, Supplementary Table 1). The $G_q$, $G_s$ and $G_{13}$ complexes have the best overall map, while the $G_i$ and $G_{12}$ complexes have relative weaker density map on helix 8 (H8) and the intracellular side of TM1-2 of receptor. We speculate those difference may reflect the coupling ability of G-proteins to receptor. Nevertheless, the stalk peptide, the ligand binding pocket and the receptor/G-protein interface are well resolved in all complexes, allowing us to obtain the structural insight into ligand engagement, receptor activation and G-protein engagement. The global structures of the GPR110/G-protein complexes resemble most of class A GPCR/G-protein complex, in which the G protein use the αH5 of Gα to engage the intracellular cavity of receptor (Fig. 1b).

## The ligand binding pocket

The receptor side is very similar when coupled with different G-proteins (Supplementary Fig. 6a). Aligned with $G_q$-coupled receptor, the overall Cα root mean squared deviation (r.m.s.d) of $G_s$-coupled, $G_i$-coupled, $G_{12}$-coupled and $G_{13}$-coupled receptor are 0.577 Å, 0.94 Å, 1.06 Å and 1.12 Å respectively. Since the $G_q$-coupled receptor has the best density map, we use the $G_q$-coupled receptor to study the stalk peptide/receptor interaction. The stalk peptide adopts twofold helix into the orthosteric ligand binding pocket formed by TM1,2,3,5,6,7 and ECL1,2,3 (Fig. 2a). The ligand binding pocket is highly hydrophobic (Supplementary Fig. 6b). Aromatic residue $W804^{6.53}$, $F747^{5.39}$, $Y668^{3.40}$, $F823^{7.42}$, $F641^{2.64}$, $W734^{ECL2}$, $W737^{ECL2}$ form the bottom and the "barrel" of the ligand binding pocket, and hydrophobic residues $L744^{5.36}$, $I811^{6.60}$, $L593^{1.47}$, $V585^{1.39}$, $V732^{ECL2}$ fill in the gap between those aromatic residues (Fig. 2c). We also find polar residues $T589^{1.43}$, $T810^{6.59}$, $R729^{ECL2}$, and $H820^{7.39}$ on the rim of the ligand binding pocket of GPR110. On the stalk peptide side, the aromatic $F569^{stalk}$ and hydrophobic $L572^{stalk}$ and $M573^{stalk}$, acts like three legs, insert deep into the hydrophobic cavity of the ligand binding pocket (Fig. 2b), make extensive hydrophobic interaction with surrounding aromatic or hydrophobic residue $F641^{2.67}$, $W734^{ECL2}$, $F747^{5.39}$, $W804^{6.53}$, $Y668^{3.40}$, $F823^{7.42}$ (Fig. 2c), particularly $F569^{stalk}$ forms a π-π interaction with $F641^{2.64}$. We summarize all ligand/receptor interactions in a connective chart (Supplementary Fig. 6d).

We used a reporter assay to validate the structural observation of stalk peptide/receptor interaction. On the stalk peptide side, we mutated the three key hydrophobic residue $F569^{stalk}$, $L572^{stalk}$ and $M573^{stalk}$ to alanine. The reporter assay data shows that those mutants decrease receptor activation activity, and double mutation of $L572^{stalk}A/M573^{stalk}A$ or triple mutation of $F569^{stalk}A/L572^{stalk}A/M573^{stalk}A$ almost completely abrogate receptor activity (Activity, Fig. 2e; protein

expression level, Supplementary Fig. 6f). Interestingly, mutations of polar residue $S568^{stalk}$ and $S570^{stalk}$ to hydrophobic leucine and alanine respectively, also decrease receptor activation activity (Activity, Fig. 2e; protein expression level, Supplementary Fig. 6f), indicating that both hydrophobic interaction and hydrophilic interaction contribute receptor binding. On the ligand binding pocket side, mutations of aromatic residue $F641^{2.64}$, $Y668^{3.40}$, $W734^{ECL2}$ and $F747^{5.39}$ of the ligand binding pocket to alanine largely decrease receptor activity in reporter assay (Activity, Fig. 2f; protein expression level, Supplementary Fig. 6f). Mutations of hydrophilic residue $T589^{1.43}A$, $R729^{ECL2}A$ and $H820^{7.39}A$ also decrease receptor interaction (Activity, Fig. 2f; protein expression level, Supplementary Fig. 6f). Of particular interest, the stalk peptide mutation $S570^{stalk}A$ and pocket mutation $H820^{7.39}A$ both significantly decrease receptor activity. Although $S570^{stalk}$ does not form a hydrogen bond with $H820^{7.39}$ in the structure, they are in close proximity of interaction. We used molecular dynamics (MD) simulation to examine whether there is an interaction between $S570^{stalk}$ and $H820^{7.39}$. A triplicated 200 ns run of MD simulation shows that the $S570^{stalk}$ and $H820^{7.39}$ do form hydrogen bond in the simulation (Supplementary Fig. 6e). Taken together, these data show that the stalk peptide engaging model of GPR110 is driven by both hydrophobic and polar interactions.

We also compared the overall ligand-engaging model of GPR110 with peptide-activated receptor ghrelin receptor[16], μ opioid receptor[23], V2 vasopressin receptor[24] and GLP1 receptor[25]. The data shows that the stalk peptide of GPR110 preferentially use the TM1 and TM7 side of the orthosteric pocket to engage receptor, while other peptides chose the middle of the ligand binding pocket for engagement. We also noticed that most of the peptide ligands (e.g. GLP1 and ghrelin) insert the head of the peptide perpendicularly into the center of the ligand binding pocket, instead of the "laydown" model of GPR110 (Fig. 2d). We also compared our GPR110 with the dopamine receptor 2 (D2R)[26] which were used in the initial model building of GPR110. The comparison shows there is a big shift of TM1 and TM7 on the extracellular side and a sharp bending on TM6 (Supplementary Fig. 6c).

## The distinctive activation conformation of GPR110

A comparison with GPR97, and the muscarinic receptor 1 (M1R), a class A GPCR, and GLP1R, a class B1 receptor, shows that the self–activated aGPCR GPR110 has a distinctive activation conformation. The most notable conformation is the deep bending of TM6 (Fig. 3a, b) in which TM6 unwinds at $6.50$ and makes a 90 degree of sharp turn at $L^{6.49}$ and then another connective 270 degrees of sharp turn at $L^{6.48}$. This sharp "bending-unwinding-turning" conformation is meditated by a "LLGL" motif in TM6 which is conserved in the aGPCR. Interestingly, the glucocorticoid-activated GPR97 does not have this sharp bending and turning conformation, indicating the tethered peptide-activated aGPCRs have a distinct activated conformation comparing to the conventional ligand-activated receptor. Interestingly, we see a similar bending and turning conformation on class B1 GLP1R[25], indicating this active conformation is evolutionary conserved in class B1 and B2 receptors.

Unlike Class A GPCRs, receptor activation is mediated by the conserved motifs such as PIF, NPxxY and DRY, no such motifs exist in aGPCR. Although class B1 receptors have a similar active conformation as in GPR110, the conserved HETX motif[27] of class B1 receptor is not conserved in aGPCRs. Since the sharp bending of TM6 directly opens the intracellular side of receptor, allowing the αH5 of Gα to engage the intracellular cavity of the receptor, we speculated this sharp bending conformation of TM6 need to be stabilize by a network interaction to maintain receptor in the open (active) conformation. We used molecular dynamics (MD) to search interactions may stabilize the active conformation. We found there is extensive hydrophobic interactions at the bending corner of TM6, specifically residue $L799^{6.48}$, $L800^{6.49}$, $M675^{3.47}$, $I630^{2.53}$ and $I834^{7.53}$ form a horizontal hydrophobic plane that

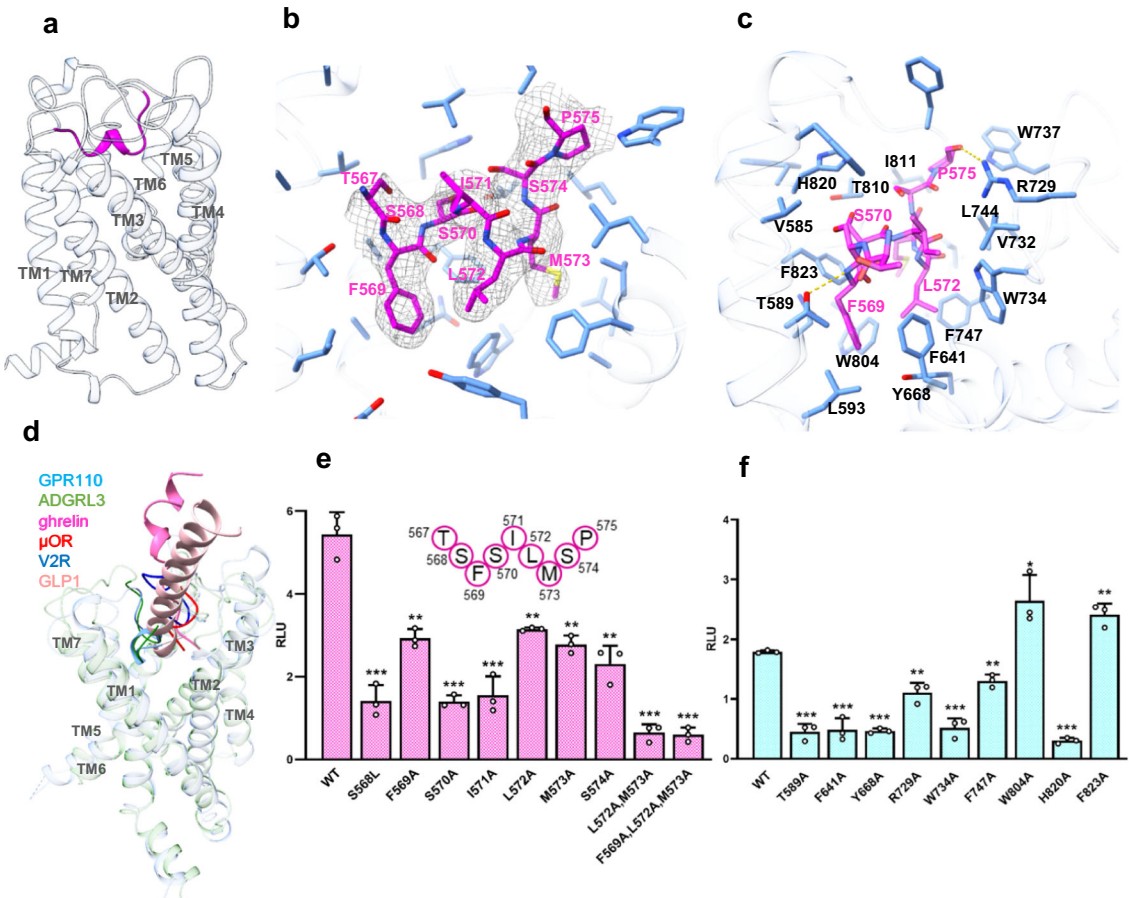

**Fig. 2 | Ligand binding pocket of GPR110. a** The overall structure of the stalk peptide (magenta color) in the ligand binding pocket of GPR110. **b** An enlarged view of the ligand binding pocket focused on the stalk peptide. Density map of the stalk peptide (blue mesh) is set at contour level of 0.08. **c** An enlarged view of the ligand binding pocket focused on the ligand binding pocket. **d** A comparison of the overall ligand engaging model of GPR110 with that of ghrelin receptor (PDB 7f9y), μOR (PDB 6dde), V2R (PDB 7dw9), GLP1R (PDB 5vai) and ADGRL3 (PDB 7sf7). **e** Mutagenesis study of stalk peptide in a NFAT-RE-Luc reporter assay. RLU, relative luciferase unit. Data are presented as mean values ± SD; *n* = 3 independent samples.

**P < 0.01; ***P < 0.001. Data between WT and mutants were analyzed by two-sided test (from left to right, P < 0.001, P = 0.0018, P < 0.001, P < 0.001, P = 0.0019, P = 0.0013, P = 0.0014, P < 0.001 and P < 0.001). Source data are provided as a Source Data file. **f** Mutagenesis study of receptor-ligand binding pocket in a NFAT-RE-Luc reporter assay. Data are presented as mean values ± SD; *n* = 3 independent samples. *P < 0.05; **P < 0.01; ***P < 0.001. Data between WT and mutants were analyzed by two-sided test (from left to right, P < 0.001, P < 0.001, P < 0.001, P = 0.0019, P < 0.001, P = 0.0014, P = 0.026, P < 0.001 and P = 0.0048). Source data are provided as a Source Data file.

stabilize the bending conformation of TM6 (Fig. 3c). We named the five residues as a "penta-core". The MD simulation data shows that the penta-core is one of the most stable conformation in the active receptor. Mutagenesis data shows that mutations of the penta-core largely decrease receptor activity in the reporter assay (Fig. 3d, Supplementary Fig. 6f).

**The G$_{12}$ and G$_{13}$ engagement**

The G$_{12}$/receptor engagement is mainly mediated by the hydrophobic interaction between αH5 of Gα$_{12}$ and TM5, TM6, TM3, TM7 and ICL1-2 of the receptor. The overall interaction is leaning toward the TM5, TM6 and TM3 side of receptor, with fewer interaction with TM7 and no connection with H8 (Fig. 4a). On the αH5 side, L367$^{G.H5.16}$, L371$^{G.H5.20}$, L376$^{G.H5.25}$, I374$^{G.H5.23}$ (for convenience of comparison we use the generic numbering of GPCR database for Gα subunit), form key hydrophobic interactions with L769$^{5.61}$, I686$^{3.58}$ and I795$^{6.44}$ (Fig. 4a). In addition, M375$^{G.H5.24}$, at the top of αH5, interact with L841$^{7.60}$. We also observe several polar interactions that positioning αH5 for receptor engagement, including the side chain of R623$^{2.42}$ forms a hydrogen bond with the carboxyl group of I374$^{G.H5.23}$, T619$^{2.45}$ forms hydrogen bond with N370$^{G.H5.19}$ and a salt bridge interaction between K616$^{ICL1}$ and E369$^{G.H5.18}$ (Fig. 4a, b). On the back view (viewing from TM3, TM4 side),

F690$^{ICL2}$ of ICL2 forms a major hydrophobic interaction with the hydrophobic pocket formed by I366$^{G.H5.15}$, V362$^{G.H5.11}$, F359$^{G.H5.08}$, I217$^{G.S3.01}$ and V41$^{G.hns.01}$, a phenomenon has been seen in many other GPCR/G-protein complexes[22,28]. Mutation of F690$^{ICL2}$ to alanine cause a dramatic decrease of receptor activity in the reporter assay, indicating the importance of this hydrophobic interaction for receptor/G$_{12}$ engagement (Supplementary Fig. 7a).

We then looked at the GPR110/G$_{13}$ engagement. The overall engagement is similar to G$_{12}$, both of them use extensive hydrophobic interactions to engage the receptor. However, notable differences are found between G$_{12}$ and G$_{13}$. For instance, in the G$_{13}$ engagement, L368$^{G.H5.16}$, L372$^{G.H5.20}$, L377$^{G.H5.25}$ line up to a stretch of leucine (Fig. 4c, brown arrow) to engage the hydrophobic groove formed by L769$^{5.61}$, V765$^{5.57}$, L796$^{6.45}$ and I795$^{6.44}$, whereas in G$_{12}$, those three leucine residues spark at different direction. Also the last residue of Gα$_{13}$, Q378$^{G.H5.26}$, are in close contact with R788$^{6.37}$. In addition, Q374$^{G.H5.22}$ forms a hydrogen bond with T619$^{2.45}$ (Fig. 4c, d). Together, those differences enable G$_{13}$ to engage receptor tighter, which explains why the GPR110/G$_{13}$ complex has a better density map and a higher resolution than the GPR110/G$_{12}$ complex. We also compared our G$_{13}$ structure with the crystal structure of Gα$_{i/13}$[29]. The comparison shows that the GPR110-bound G$_{13}$ structure matches very well with the Ras-like

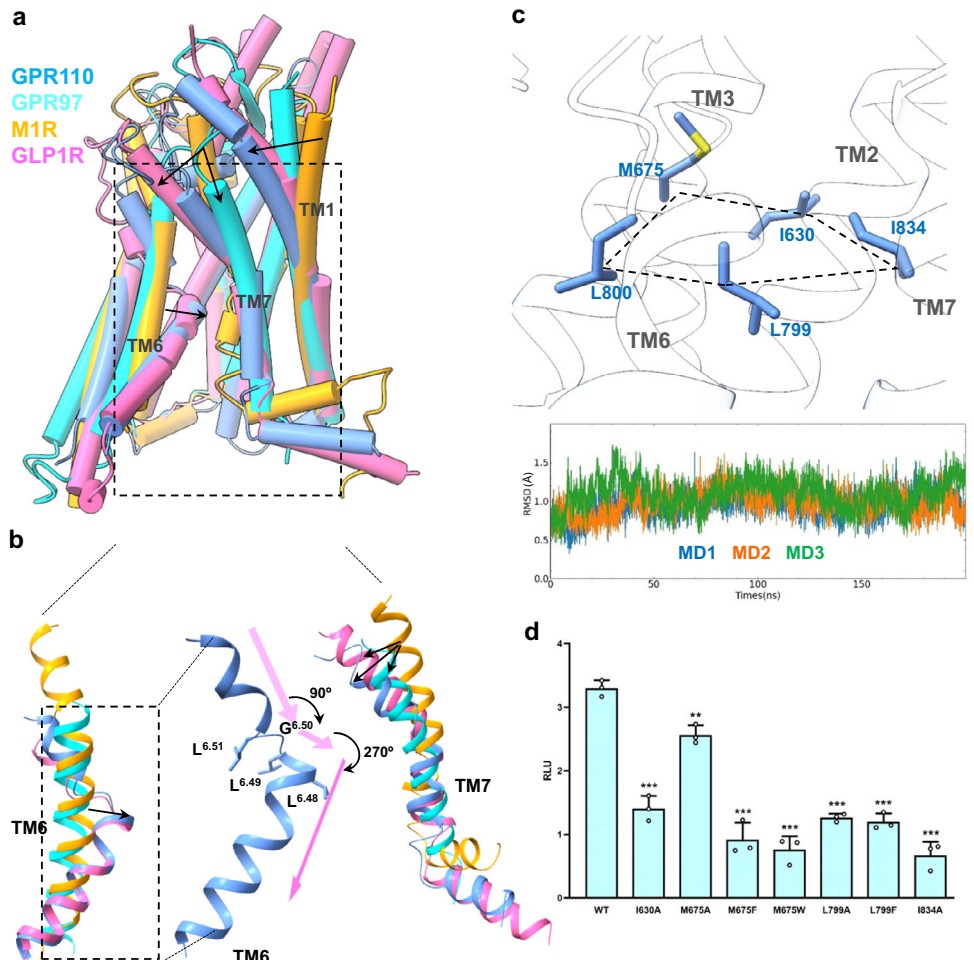

**Fig. 3 | A distinct active conformation of GPR110. a** A comparison of the overall active conformation of GPR110 with M1R (PDB 6oij), GPR97 (PDB 7d77) and GLP1R (PDB 5vai). **b** A detailed comparison of TM6 and TM7 in the active conformations. **c** The "penta-core" of GPR110. Upper panel, the penta-core; lower panel, trajectory analysis of the penta-core of GPR110 in MD simulations. **d** Mutagenesis study of the penta-core of GPR110 in a NFAT-RE-Luc reporter assay. RLU, relative luciferase unit. Data are presented as mean values ± SD; $n = 3$ independent samples. $**P < 0.01$; $***P < 0.001$. Data between WT and mutants were analyzed by two-sided test (from left to right, $P < 0.001$, $P = 0.0029$, $P < 0.001$, $P < 0.001$, $P < 0.001$, $P < 0.001$ and $P < 0.001$). Source data are provided as a Source Data file.

domain of the crystal structure with an overall of 0.899 Å r.m.s.d over 199 pairs of Cα (Supplementary Fig. 7b).

## A comparison of $G_q$, $G_s$, $G_i$, $G_{12}$ and $G_{13}$ engagements in one single receptor

With the GPR110/$G_q$, GPR110/$G_s$, GPR110/$G_i$, GPR110/$G_{12}$ and GPR110/$G_{13}$ complex structures, we were able to systematically compare all major G-protein couplings to the same receptor. The receptor/G-protein engagement is mainly mediated by the interaction between the distal region of αH5 and the intracellular cavity of receptor. A comparison of the overall αH5 engagements shows that distances between αH5 and the TM3/TM5 groove are $G_{13} < G_{12} < G_s, < G_q < G_i$, with αH5 of $G_{13}$ closet to the TM3/TM5 groove and $G_i$ farthest to the TM3/TM5 groove (Fig. 5a). The overall receptor interface for $G_q$, $G_s$, $G_i$, $G_{12}$ and $G_{13}$ are 991, 1020, 853, 1056, and 988 Å², respectively. It is generally accepted that G-protein selectivity is largely determined by the last 7 residues of the αH5 of Gα subunit. We align the protein sequence of $G\alpha_q$, $G\alpha_s$, $G\alpha_i$, $G\alpha_{12}$ and $G\alpha_{13}$, and renumber the C-termini of αH5 in a reverse order (start from −1 at the C-terminus) for easy comparison (Fig. 5b, right panel). A superimposition of the C-termini of αH5 of GPR110-bound $G\alpha_q$, $G\alpha_s$, $G\alpha_i$, $G\alpha_{12}$ and $G\alpha_{13}$ shows a clear classification between $G\alpha_q/G\alpha_s$ and $G\alpha_i/G\alpha_{12}/G\alpha_{13}$ (Fig. 5b). The most notable feature is the polar residues at position −3 and −4 that distinguish $G\alpha_q/G\alpha_s$ from $G\alpha_i/G\alpha_{12}/G\alpha_{13}$. Particularly, position −4 is a bulky tyrosine for $G\alpha_q$

and $G\alpha_s$, while it is a hydrophobic residue (C, I, L) for $G\alpha_i$, $G\alpha_{12}$ and $G\alpha_{13}$ (Fig. 5b). At position −3, $G\alpha_q$ and $G\alpha_s$ have a polar residue N and E respectively, $G\alpha_{12}$ and $G\alpha_{13}$ have a hydrophobic methionine, and $G\alpha_i$ has a glycine. In positon −5, all the 5 G-proteins have a polar residue. Based on the above observation, we classify G-protein engagements into 2 major classes, Class I, $G\alpha_q/G\alpha_s$ engagement; Class II, $G\alpha_i/G\alpha_{12}/G\alpha_{13}$. In Class I, the engaging model of $G_q$ and $G_s$ are very similar. The Y at position −4 of $G\alpha_q$ and $G\alpha_s$ all point to the gap between TM3 and TM2, surrounding by $R623^{2.46}$, $L682^{3.54}$, $L681^{3.53}$ and $R685^{3.57}$ (Fig. 5c). There is a subtle difference between $G_q$ and $G_s$ in the polar interaction of position −3, in which the $N^{-3}$ of $G\alpha_q$ interacts with the D842 of the receptor and $E^{-3}$ of $G\alpha_s$ interacts with $S843^{8.48}$ of the receptor (Fig. 5c).

In Class II, hydrophobic interactions are the driven force for $G_i$, $G_{12}$ and $G_{13}$ engagements as the C-ends of αH5 of $G\alpha_i$, $G\alpha_{12}$ and $G\alpha_{13}$ are most hydrophobic, particularly, hydrophobic residues of position −2 and −4 to make extensive hydrophobic interactions with TM5, TM6 and TM3, including $V765^{5.57}$, $L796^{6.45}$ and $L681^{3.53}$. Interestingly, $R623^{2.46}$ forms hydrogen bonds with the backbone carboxyl group of position −4 of $G\alpha_i$, $G\alpha_{12}$ and $G\alpha_{13}$. We also observed a substantial difference between $G_i$ and $G_{12}/G_{13}$ engagements. $G_{12}$ and $G_{13}$ have more polar interactions than $G_i$ in the middle region of αH5 (Fig. 5d), specifically, $D^{-5}$ of $G\alpha_{13}$ interacts with $T619^{2.42}$ and $E^{-9}$ of $G\alpha_{12}$ interacts with $K616^{ICL1}$, and this may explain why $G_{12}$ and $G_{13}$ show a better coupling than $G_i$ in GPR110.

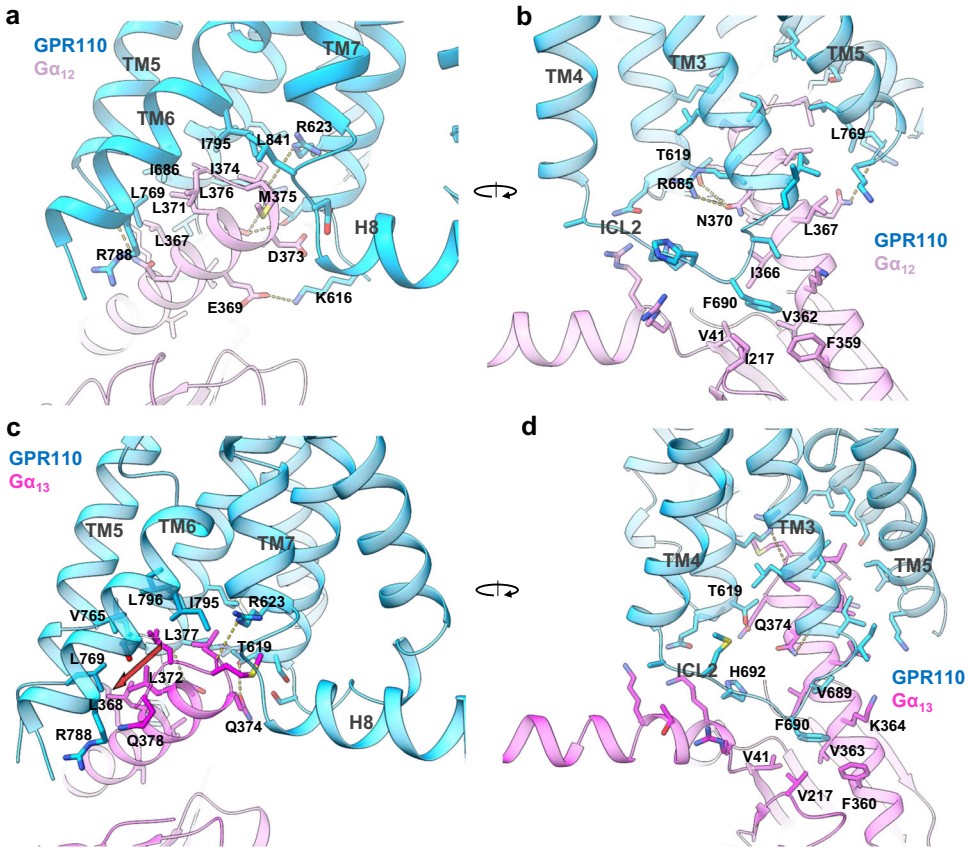

**Fig. 4 | The G₁₂ G₁₃ engagements of GPR110. a, b** The engagement of G₁₂ to GPR110, viewing from TM6/TM7 side and TM4/TM3 side, respectively. **c, d** The engagement of G₁₃ to GPR110, viewing from TM6/TM7 side and TM4/TM3 side, respectively.

The biggest difference between the $G_q/G_s$ and the $G_i/G_{12}/G_{13}$ engagements is the participation of hydrogen bond in the engagements. The tyrosine at −4 position and glutamate or asparagine at the −3 positon enable polar interaction with receptor, for instance $N^{-3}$ of $G\alpha_q$ interact with $D842^{8.47}$, and $E^{-3}$ of Gαs interact with $S843^{8.48}$ (Fig. 5c). We count all interactions of the distal region of αH5 of all G alpha subunits with the receptor in a connective map (Fig. 5e), the data shows there are more hydrogen bond interactions in the $G_q$ and $G_s$ engagement than in the $G_i$, $G_{12}$ and $G_{13}$ engagement. We also saw more hydrophobic interactions in the $G_{12}/G_{13}$ engagement than the $G_i$ engagement. In fact, the $G_i$ complex has fewest interactions (hydrophobic and polar), and this explains the poor coupling of $G_i$ to the receptor.

**Physiological relevance**

In this study, we used reporter assay to demonstrate that GPR110 is a pluripotent receptor capable of coupling to $G_q$, $G_s$, $G_i$, $G_{12/13}$ proteins. Since only $G_s$ and $G_i$ coupling activities of GPR110 were reported before, we asked whether GPR110 can couple to $G_q$ and $G_{12/13}$ proteins in a more physiological setting. We therefore adopted a newly developed bioluminescence resonance energy transfer (BRET) assay which can quantitatively measure constitutive activity of GPCR to assess the intrinsic activity of GPR110 CTF. The data shows that GPR110 is able to couple to all 4 major G-proteins, interestingly, the BRET assay data also shows that $G_q$, $G_{13}$ and $G_s$ have strong coupling activity to GPR110, while $G_i$ has the weakest coupling activity (Fig. 6a), consistent with our structural observation that $G_q$, $G_{13}$ and $G_s$ complexes have a strong receptor/G-protein association and a high quality density map, while the $G_i$ complex has the poorest receptor/G-protein association and density map.

A previous study of GPR110 identified synaptamide, a metabolite of DHA, as a ligand of GPR110[8]. A following study suggested that

synaptamide bound to the GAIN domain and the receptor was activated by the interaction between GAIN domain and the transmembrane domain of receptor, but not by the stalk peptide[10]. However, our structures of the activated GPR110 clearly show that the receptor is activated by the tethered-stalk peptide mechanism. We therefore asked whether the adding of stalk peptide (TSFSILMSPN, 567-576) to the primary culture of neuron isolated from mouse brain will have similar effect as DHA. The data shows that the adding of stalk peptide to the culture promotes neurite outgrowth, similar to the effect of DHA (Fig. 6b, c), suggesting that GPR110 is activated through the tethered-stalk peptide mechanism under physiological condition.

**Discussion**

In this study, we demonstrated that GPR110 is a pluripotent GPCR that is able to couple all 4 major G-protein pathways and forms complexes with $G_q$, $G_s$, $G_i$, $G_{12}$ and $G_{13}$. Those properties enable us to obtain structural information of GPR110 in complexes with $G_q$, $G_s$, $G_i$, $G_{12}$ and $G_{13}$. The direct comparisons of $G_q$, $G_s$, $G_i$, $G_{12}$ and $G_{13}$ with the same receptor yielded the most insightful structural information about G-protein coupling selectivity. For instance, we identified positon −4 of the αH5 of Gα as a dividing point that separates $G_q/G_s$ engagements (Class I) from $G_i/G_{12}/G_{13}$ engagements (Class II) based on hydrophobic and hydrophilic residue distribution at this key position. In this case, $G_q$ and $G_s$ both have tyrosine at position −4, while $G_i/G_{12}/G_{13}$ have a small hydrophobic residue, indicating $G_q/G_s$ use a similar pattern to couple receptor while $G_i/G_{12}/G_{13}$ use a different tactic. Several structural studies of aGPCR have been recently been published[30–33], one of those studies revealed structures of ADGRF1 in complex with $G_s$ and $G_i$[33]. A comparison of our study with those published works reveal similar mechanism of the self-activation by tethered agonist of aGPCR. A superimposition of our

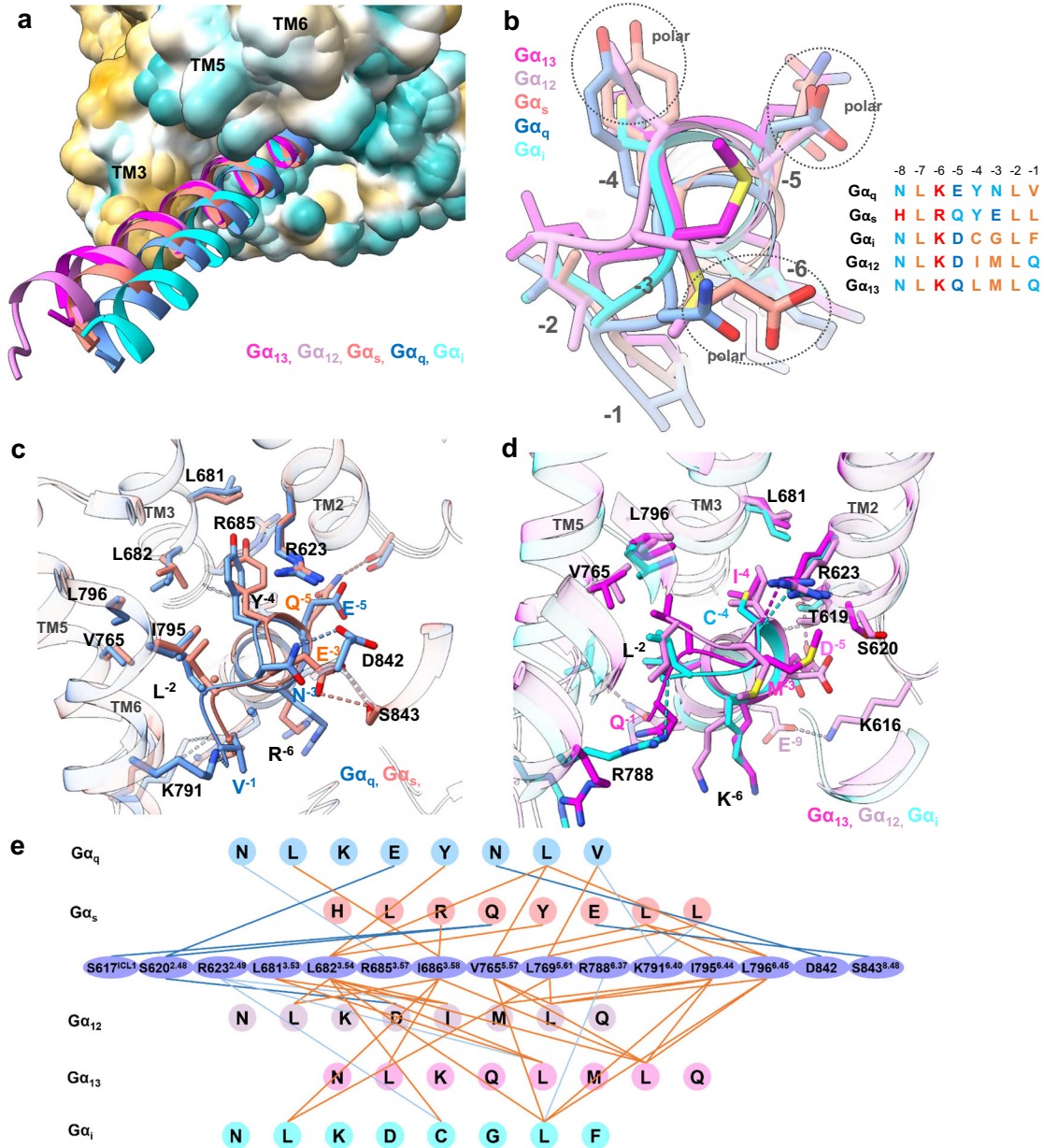

**Fig. 5 | A comparison of GPR110 engagements with all 4 major G-proteins. a** An overall comparison of αH5 engagements of $G\alpha_q$, $G\alpha_s$, $G\alpha_i$, $G\alpha_{12}$ and $G\alpha_{13}$ to receptor. Receptor was drawn in hydrophobic surface potential. **b** A comparison of the very end of αH5 of G-proteins in engagements of GPR110. Left panel, superimposition of αH5 of $G\alpha_q$, $G\alpha_s$, $G\alpha_i$, $G\alpha_{12}$ and $G\alpha_{13}$ in GPR110 engagements; right panel, an alignment of the last 8 residues of αH5 of $G\alpha_q$, $G\alpha_s$, $G\alpha_i$, $G\alpha_{12}$ and $G\alpha_{13}$. Brown color marks hydrophobic residues, cyan color marks polar residues, blue color marks negative charged residues and red color marks positive charged residues. **c** A comparison of the $G_q$ and $G_s$ engagements with GPR110. **d** A comparison of the $G_i$, $G_{13}$ and $G_{12}$ engagements with GPR110. **e** A connective interaction map of the αH5/receptor interaction of the $G\alpha_q$, $G\alpha_s$, $G\alpha_i$, $G\alpha_{12}$, $G\alpha_{13}$/receptor complexes. The thick blue line marks hydrogen bond between side chains, the light and thin blue line marks hydrogen bond between side chain and backbone, the brown line marks hydrophobic interaction.

GPR110/$G_s$ and GPR110/$G_i$ complexes with the ADGRF1/$G_s$ and ADGRF1/$G_i$ complexes[33] shows the structures are very similar with a r.m.s.d of 0.919 Å and 1.089 Å for $G_s$ and $G_i$ complex respectively (Supplementary Fig. 8). In addition, we revealed GPR110 in complex with $G_q$, $G_{12}$ and $G_{13}$, and more importantly, compared all major G-protein ($G_q$, $G_s$, $G_i$, $G_{12}$ and $G_{13}$) couplings with the same receptor and the comparison, providing pivotal information on G-protein. We further demonstrated GPR110 is able to couple to 4 major G-protein signaling in a more physiological setting, and provided direct evidence that GPR110 is activated through the tethered stalk peptide mechanism physiologically. Together with the detailed analysis of the stalk peptide binding, our study provides a framework for understanding aGPCR activation and GPR110 signaling.

## Methods

### Constructs

The codon-optimized human GPR110 gene (residues 567-873) was fused with a LgBiT fusion to its C-terminus, followed by a Tobacco etch virus (TEV) cutting site and 2 x maltose-binding protein (MBP) was cloned in pFastBac1 baculovirus expression vector. The C-terminus HiBiT fusion of human $G\beta_1$ was cloned into pFastBac plasmid as the VIP1R paper[15]. The mini-$G\alpha_q$ and mini-$G\alpha_s$ constructs are adopted from the ghrelin receptor/$G_q$ complex paper[16] and the melanocortin receptor 1/$G_s$ complex paper[17], respectively, the sequences were codon-optimized and synthesized by Langjing Biotech, Shanghai, and inserted into pFastBac. For $G\alpha_{12}$, the "GGSGG" linker of mini-$G\alpha_{12}$8 was swapped with the $G\alpha_i$ AHD domain, and the first 19 residues of mini-

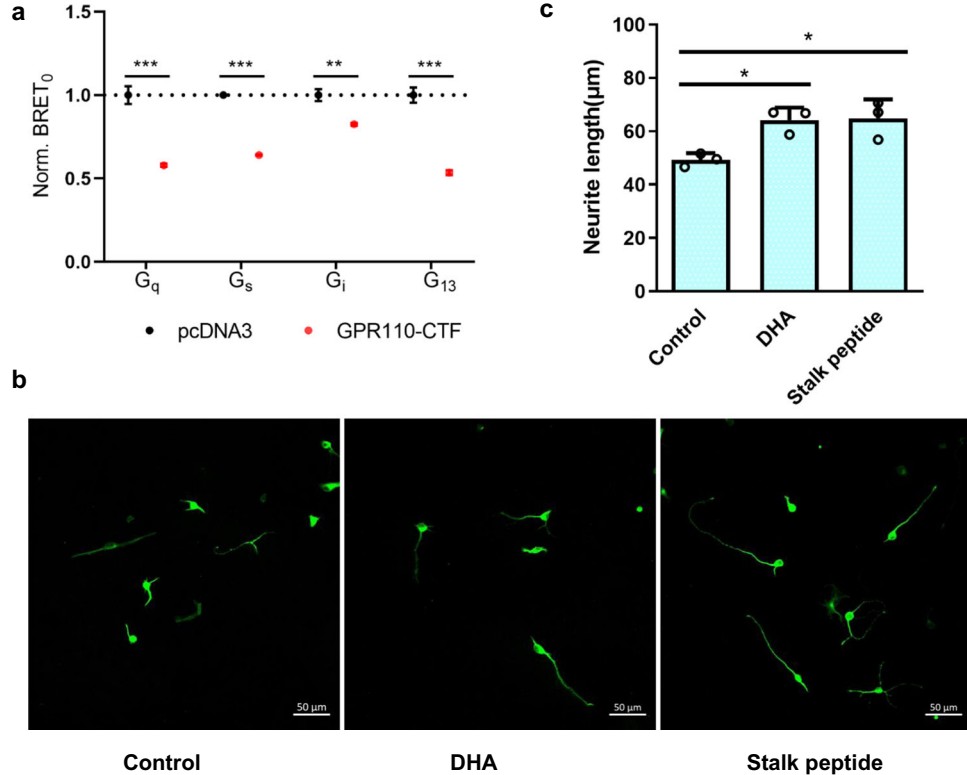

**Fig. 6 | Physiological relevance of the G-protein coupling of GPR110. a** A BRET assay to examine the constitutive activity of GPR110 CTF. Data are presented as mean values ± SD; $n = 3$ independent samples. **$P < 0.01$; ***$P < 0.001$. Data between WT and mutants were analyzed by two-sided test (for $G_q$ group, $P < 0.001$; for $G_s$ group, $P < 0.001$; for $G_i$ group, $P = 0.0012$; for $G_{13}$ group, $P < 0.001$). Source data are provided as a Source Data file. **b** The stalk peptide (100 μM) of GPR110 increases neurite outgrowth. **c** Measurement of neurite outgrowth. $n = 3$, 20 cells were analyzed per well, *$P < 0.05$. All data were analyzed by two-sided, one-way ANOVA with Tukey's test (from left to right, $P = 0.0233$, $P = 0.0194$). Source data are provided as a Source Data file.

$G\alpha_{12}$ was substituted with the first 18 residues of $G\alpha_i$, again the sequence was codon-optimized and synthesized by Langjing Biotech, Shanghai, and inserted into pFastBac1. Similar strategy was used in the $G\alpha_{13}$ construct in which AHD of $G\alpha_{13}$ was swapped with AHD of $G\alpha_i$ and the first 19 residues of $G\alpha_{13}$ was replaced with the first 18 residues of $G\alpha_i$ (Supplementary Fig. 2). For $G\alpha_i$, we use a dominant-negative human $G\alpha_{i1}$ (G203A, A326S) as before[18]. The wild-type human $G\beta_1$, wild-type human $G\gamma_2$ were cloned into pFastBac 1 plasmid. The scFv16 encoding the single-chain variable fragment of mAb16 as described before[34].

### Expression and purification of GPR110/G-protein complexes
For $G_s$ and $G_q$ complex assembling, baculovirus encoding the GPR110, $G\alpha$ (mini-$G\alpha_s$ or mini-$G\alpha_q$), $G\beta_1$, and $G\gamma_2$ were co-infected into the Spodoptera frugiperda (Sf9) cells ($2 \times 10^6$ cells per ml) at a ratio of 1:100 (virus volume vs cells volume) and cells were harvested 48 h postinfection. For $G_i$, $G_{12}$ and $G_{13}$ complex assembling, baculovirus encoding the GPR110, $G\alpha$, $G\beta_1$, $G\gamma_2$ and scFv16 were co-infected into the Spodoptera frugiperda (Sf9) cells ($2 \times 10^6$ cells per ml) at a ratio of 1:100 (virus volume vs cells volume) and cells were harvested 48 h postinfection. Cell pellets were resuspended in 20 mM Hepes buffer, 150 mM NaCl, 10 mM MgCl₂, 20 mM KCl, 5 mM CaCl₂, pH 7.5, with 0.5 mU/ml apyrase and homogenized by douncing ~30 times. After 1 h incubation of the lysis at room temperature, 0.5% (w/v) lauryl maltose neopentylglycol (LMNG, Anatrace), 0.1% (w/v) cholesteryl hemisuccinate TRIS salt (CHS) were added to solubilize the membrane at 4 °C for 2 h. Then the lysis was ultracentrifuged at 65,000 g at 4 °C for 40 min. The supernatant was incubated with amylose column for 2 h then washed with a buffer of 20 mM Hepes, pH 7.5, 150 mM NaCl and

0.01% LMNG/0.002% CHS, and eluted with the same buffer plus 10 mM maltose. The elution was concentrated and cut with home-made TEV overnight at 4 °C, then separated on a Superdex 200 Increase 10/300 GL (GE health science) gel infiltration column with a buffer of 20 mM Hepes, pH 7.5, 150 mM NaCl and 0.00075% (w/v) LMNG, 0.00025% glyco-diosgenin (GDN), 0.0002% (w/v) CHS (Anatrace). The GPR110/G-protein complex corresponding peak was concentrated at about 10 mg/ml and snap frozen for later cryo-EM grid preparation.

### Expression and purification of Nb35
Nanobody-35 (Nb35)[22] bearing a C-terminal His-tag was expressed in the periplasm of E. coli BL21, and grown in a TB culture medium with 100 μg/mL ampicillin, 2 mM MgCl₂, and 0.1% (w/v) glucose at 37 °C, 200 rpm. When OD600 reached 0.7–0.9, 1 mM IPTG was added to induce its expression. Induced cultures were grown 4-6 h at 28 °C. The cells were collected by centrifugation and lysed in a frozen buffer solution (50 mM Tris pH 8.0, 0.125 mM sucrose, 0.125 mM EDTA). After lysis, cell fragments were removed by centrifugation and Nb35 was purified by nickel affinity chromatography. The purified Nb35 was added with 10% (V/V) glycerol and stored at 80°C for use.

### Grid preparation and cryo-EM data collection
A 3 μl receptor/G-protein complex sample (~10 mg/ml) was applied to a glow-charged quantifoil R1.2/1.3 Cu holey carbon grids (Quantifoil GmbH). The grids were vitrified in liquid ethane on a Vitrobot Mark IV (Thermo Fisher Scientific) instrument at setting of blot force of 10, blot time of 5 s, humidity of 100%, temperature of 4 °C. Grids were first screened on a FEI 200 kV Arctica transmission electron microscope (TEM) and then grids with evenly distributed particles in thin ice were

transferred to a FEI 300 kV Titan Krios TEM equipped with a Gatan Quantum energy filter. Images were taken by a Gatan K2 direct electron detector at magnitude of 64,000, super-resolution counting model at pixel size of 0.55 Å, the energy filter slit was set to 20 eV. Each image was dose-fractionated in 40 frames using a total exposure time of 7.3 second at a dose rate of 1.5 e/Å$^2$/s (total dose 60 e/Å$^2$). All image stacks were collected by the FEI EPU program, nominal defocus value varied from −1.2 to −2.2 μm.

## Data processing

We use same pipeline to process data as described before. For the $G_q$, $G_s$ and $G_i$ complex, a total of 2000–2300 raw movies (0.55 Å) were binned (1.1 Å) and motion-corrected using MotionCor2[35]; for the $G_{12}$ and $G_{13}$ complex, a total of 3200–34000 raw movies (0.54 Å) were binned (1.08 Å) and motion-corrected using MotionCor2. Then the motion-corrected movies were processed by CTF estimation by CTFFIND 4.1[36]. Particles (-1.5–4.0 million) were picked by crYOLO[37] and extracted by RELION[38] (version 3.1) and subjected to reference-free 2D classification in RELION. Good classes (-0.7–1.3 million particles) which of well-defined features were passed to next round for initial model generation and 3D classification. The initial model was generated by cryoSPARC[39] ab initio. The model was used as reference in RELION 3D classification (-5 classes). The best class (-500,000) that showed clear secondary structure features was selected for a 3D refinement in RELION, followed by a Baysian polishing[40], then a 3D refinement and a CTF refinement in RELION. The refined particles were subjected to a second round 3D classification (3–4 classed) with mask on the complex to yield a class of about 260,000–500,000 particles for final refinement by the cryoSPARC Non-uniform Refinement, which generated a map of 2.66–3.09 Å, based on the gold standard Fourier Shell Correlation (FSC) = 0.143 criterion. Local resolution estimations were performed using an implemented program in cryoSPARC. The final map was post-processed by DeepEMhancer[41].

## Model building

The AlphaFold[42] structures of human GPR110 (AF-Q5T610-F1) and the $G_i$ protein complex from the D2R (PDB 6vms and 7jvr)[26,43] or Gs protein complex from M1R (PDB 7f4d)[17] were used as initial models for model rebuilding and refinement against the electron microscopy map. All models were docked into the electron microscopy density map using UCSF Chimera[44] then subjected to iterative manual adjustment in Coot[45], followed by a rosetta cryoEM refinement[46] at relax model and Phenix real space refinement[47]. The model statistics were validated using MolProbity[48]. Structural Figures were prepared in UCSF Chimera, ChimeraX[49] and PyMOL (https://pymol.org/2/).

## Structure and sequence comparison

Sequence alignment by the Clustal Omega[50] sever and the representation of sequence alignment was generated using the ESPript[51] website (http://espript.ibcp.fr). The generic residue numbering of GPCR is based on the GPCRdb[52] (https://gpcrdb.org/).

## The cell based reporter assays

The SRE, CRE, SRF-RE and NFAT-RE reporter assays (Promega) were performed by the Promega instruction as described before[14,34]. Briefly, AD293 cells were split into 24 well plate at a density of 40,000 per well then transfected with 100 ng of SRE-Luc (or CRE-Luc, or SRF-RE-Luc, or NFAT-RE-Luc) 10 ng of pcDNA3-GPR110 wild-type or mutations, 10 ng of phRGtkRenilla plasmids (Promega) by X-tremeGENE HP (Roche) at a ratio 3:1 to DNA amount after one day of growth on 37 °C at 5% $CO_2$. 24 h after transfection, cells were harvested and lysed by addition of 1× Passive Lysis Buffer (Promega). The luciferase activity was assessed by the Dual-Glo Luciferase system (Promega). Data were plotted as firefly luciferase activity normalized to Renilla luciferase activity in Relative Luciferase Units (RLU).

## Animals

Pregnant female C57BL/6 mice were obtained from Charles River Laboratories (Beijing, China) for the preparation of primary cortical neurons. Animal experiments were carried out in strict accordance with the Guide for the Care and Use of Laboratory Animals (8th edition) and approved by the Institutional Animal Care and Use Committee of Harbin Institute of Technology (HIT/IACUC).

## Primary cell culture and treatment

Primary cortical neurons were prepared using a previously described method[53]. Cortices were isolated from P1 pups and digested with 0.25% trypsin (Solarbio, T1350) for 20 min at 37°C. The digestion was terminated by the addition of 10% foetal bovine serum (FBS, ExCell, FND025) in DMEM medium (Hyclone, SH30022.01) and mechanically disrupted by pipetting several times to make a homogenous mixture, which was passed through a cell strainer (BD Falcon, 352350) to remove undissociated tissue. The cells were centrifuged for 3 min at 1200 g and resuspended in DMEM supplemented with 10% FBS (ExCell, FND025) and 1% penicillin-streptomycin (Gibco, 15140). The dissociated cortical neurons were seeded in poly-D-lysine-coated (Sigma, P4707) chamber slides in 24-well plates (2.5 × 10$^4$ cells per well) for neurite outgrowth analysis. After 4 h, the medium was changed to neurobasal medium (Gibco, 21103) supplemented with 2% B27 (Invitrogen, 17504044), 1% L(+)-Glutamine (Gibco, 25030) and 1% penicillin-streptomycin (Gibco, 15140). Cortical neurons were treated with 1 μM DHA, 100 μM stalk-peptide or vehicle to each group separately on day 1 in vitro for 24 h.

## Immunocytochemistry

For immunofluorescence, cells plated on chamber slides were fixed with 4% Paraformaldehyde at room temperature for 15 min. Following washing three times with PBS, cells were blocked by PBS containing 0.2% Triton X-100 (Vetec, V900502) and 10% goat serum for 1 h. Then cells were incubated in primary antibody solution overnight at 4°C. The primary antibody used was rabbit anti-MAP2 antibody (1:2000, Abcam, ab281588). After three times washing with PBS, cells were incubated with donkey anti-rabbit secondary antibody (1:1000, Abcam, ab150073) for 1 h at room temperature. After washing the secondary antibody, DAPI (Alphabio, A1013) was added in the chamber for 10 min to stain the cell nuclei. The cells were examined with laser scanning confocal microscope (Zeiss, LSM880). Neurite outgrowth was evaluated for a total of 60-120 neurons per group using the Image J software, by taking 5 images containing 4-8 MAP2-positive neurons per image from triplicate samples in each of three independent experiments.

## BRET assay using tricistronic activity sensors

The tricistronic activity sensors assay was performed as previously described[54]. The G-protein sensor plasmids was obtained from Addgene (https://addgene.org/Gunnar_Schulte/). For measuring constitutive activity, 500 ng of GPCR was co-transfected with 500 ng of G protein sensor into AD293 cells by Lipofectamine 2000 (Thermo Fisher Scientific). After incubation for 24 h in a 37 °C, 5% $CO_2$ atmosphere, the transfected cells were seeded into 96-well plates and incubated for another 24 h. Cells were washed with HBSS, and incubated with 1:1000 dilution of furimazine stock solution. The BRET ratio was measured in three consecutive reads after incubation for 3 min at 37 °C using EnVision multimode plate reader (PerkinElmer). The signal was calculated as the ratio of 460/40-nm monochromator (gain, 3600) and cpVenus emission using a 535/30-nm monochromator (gain, 4000) with an integration time of 0.3 s in both channels, according to previous report.

## Western-Blot

AD293 cells were transfected with 100 ng pcDNA3-GPR110 per well of 24 well plate by PEI at the ratio of 1:5. Two days after transfection, cells

were lysed by cell lytic reagent (Sigma), proteins in cell lysates were separated in 10% Bis-Tris gels at 170 V for 1 h and then transferred onto polyvinylidene difluoride (PVDF) membranes at 100 V for 1.5 h. The membranes were blocked with 10% milk in TBS-T (20 mM Tris-HCl, pH 7.5, 50 mM NaCl, 0.1% Tween-20) at room temperature for 30 min. One of the membranes was incubated at room temperature for 2 h with monoclonal anti-FLAG M2-peroxidase (HRP) antibody (1:5000, Sigma) in TBS-T. The other one was incubated with β-actin mouse mAb (1:10,000, ABclonal) in TBS-T containg 3% milk at room temperature for 2 h, and after being washed with TBS-T, the membrane was incubated for 30 min with HRP goat anti-mouse IgG (1:5000, ABclonal) in TBS-T. After treating with chemiluminescent substrate (Thermo Fisher Scientific), protein bands were detected by iBright CL1000 imaging system (Thermo Fisher Scientific).

### Molecular dynamics simulation

The cryo-EM structure of GPR110 (receptor only) was used to initial model in the MD simulation. The ICL3 break (774-785) was filled with 4 alanine residues. Using CHARMM-GUI[55,56], the receptor was inserted into a bilayer lipid contain POPC (palmitoyl-2-oleoyl-sn-glycero-3-phosphocholine) and cholesterol at ratio of 4:1, the membrane size is 65 × 65 Å with 22.5 Å water and ion 0.15 M KCl in the top and bottom, temperature 303.15 K. The Amber force fields were set to: protein FF19SB, lipid LIPID17, and water TIP3P. The simulations were performed by Amber20 package[57]. The system was first energy minimized for solvent and all atoms, heat to 300 K in 300 ps and then equilibrated for 700 ps, followed by three independent production runs of 200 ns with a timestep of 2 fs. During simulations, Particle mesh Ewald algorithm were applied for the calculation of long-range electrostatic interaction and a cutoff of 10 Å were applied for short-range electrostatic interaction and van der Waals interactions. All bonds with hydrogens are constrained by SHAKE algorithm. The system temperature (300 K) and pressure (1 atm) were controlled by Langevin thermostat and Berendsen barostat, respectively. The trajectories were analyzed and visualized in VMD[58].

### Reporting summary

Further information on research design is available in the Nature Research Reporting Summary linked to this article.

## Data availability

The data that support this study are available from the corresponding authors upon request. The cryo-EM density maps and atomic coordinates have been deposited in the Electron Microscopy Data Bank (EMDB) and Protein Data Bank (PDB) under accession numbers EMD-32881 and 7WXU for the GPR110/G$_q$ complex; EMD-32882 and 7WXW for the GPR110/G$_s$ complex; EMD-32972 and 7X2V for the GPR110/G$_i$ complex; EMD-32905 and 7WZ7 for the GPR110/G$_{12}$ complex, EMD-32883 and 7WY0 for the GPR110/G$_{13}$ complex, respectively. Source data are provided with this paper.

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

## Acknowledgements

We thank Dr. Zhiwei Huang of the HIT Center for Life Sciences for suggestion and support of this project. This work was supported by the National Natural Science Foundation of China (32070048 to Y.H.).

## Author contributions

X.Z. made the expression constructs, purified the proteins, prepared and screened the grids, made the mutations, performed functional assays and analyzed data. Y.Q. made mutations, purified proteins and performed functional assays. X.L. performs functional assays. R.X., Z.X., N.W., J.L., and H.Y. prepared plasmids, cultured cells and prepared reagent. A.Z., and C.G. set cryo-EM and collected data. G.W. edit the manuscript and supervised functional study. Y.H. conceived the project, designed the experiments, analyzed data, collected data, solved the structures, performed MD, wrote the manuscript and supervised the project. All authors contributed to data interpretation and preparation of the manuscript.

## Competing interests

The authors declare no competing interests.
