## [Peer Review File · Nature Communications]

REVIEWER COMMENTS

Reviewer #1 (Remarks to the Author):

In this study, Zhu and colleagues determined five structures of adhesion receptor GPR110 in complex with different types of G proteins by cryo-EM. Despite several adhesion receptors have been published on Nature (doi: 10.1038/s41586-022-04619-y, doi: 10.1038/s41586-022-04590-8), this study firstly reported a series of G protein bound activated GPR110 and compared the differences of molecular mechanism of G protein coupling. Structural comparison reveals activation mechanism of GPR110, as well as two classes of G protein subunits in GPR110 complexes. This study is important for structure and pharmacology of GPCR field, but some concerns should be addressed in the future revised version.

1. GPR110 exerts its multi-pharmacological functions through different G protein subunits, can the author provide more evidence to elucidate primary signaling pathway of GPR110? Can the author carry out more mutations in ligand binding pocket and test their contribution for different G proteins signaling? If so, could the author design new peptide for biased signaling pathway of GPR110? This is one of the excellent points of this study, I appreciate if the author can offer additional information.
2. The author should generate more mutations to validate the different G protein coupling mode.
3. Could the author test the expression level of mutations of GPR110?

Reviewer #2 (Remarks to the Author):

This paper reports the first set of structures of an adhesion GPCR, GPR110, activated by its tether stalk motif and its coupling with five different subtypes of G proteins, Gq, Gs, Gi, G12 and G13. The structures are of high quality with appropriate resolutions, which reveal clear feature of the stalk motif and G protein coupling interactions. The structure information is well supported by mutational studies related to the stalk motif binding pocket and the G protein couple interface. The manuscript is very well written and the figures are nicely prepared. A similar structure has been published online by Nature recently but this current paper reports the work well beyond what has been reported in Nature. I would recommend its publication ASAP to catch what have been published in Nature this Month, after the minor points below have been addressed.

1. The title should include adhesion GPCR and stalk motif.

2. The common name of adhesion GPCR for GPR110 should also be in the abstract.
3. The original conserved HTEX motif in class B GPCRs was reported by Yin et al ([https://www.jbc.org/article/S0021-9258\(20\)40795-1/fulltext](https://www.jbc.org/article/S0021-9258(20)40795-1/fulltext)), and the correct citation should be used.
4. In reference 40, the authors used the Dopamine receptor D2R as the initial model for their model building. However, this model has poor resolution and density in the TM1 region of the receptor. The correct model of D2R should be 7JVR, which has better resolution with clear density for TM1. It will be good to compare the two D2R structure with GPR110.

Reviewer #3 (Remarks to the Author):

I have no major concerns about the manuscript. Indeed, data for G12/13 in PDB is highly limited comparing Gs or Gi. This confirms that the manuscript includes outstanding findings and the suggestion stated in lines 129-130 even goes far beyond when joined with line 139.

Some minor points are listed below:

- “self-tethering” activation – instead: ‘self-activation by tethered agonist’ or ‘self-cleavage to release tethered agonist’ or ‘tethered self-activation’ would be better.
- ‘The autoproteolysis also results a split of the receptor into two fragments, the N-terminal fragment (NTF) and C-terminal fragment (CTF). The CTF contains the stalk peptide,’ instead: ‘The autoproteolysis allows to release of the N-terminal fragment (NTF) of the receptor. The remaining N-terminal region of the receptor includes the stalk peptide.’. Using CTF in this case may unnecessarily suggest that the tethered agonist is formed from the C-terminal region of the receptor and not from its N-terminal region, located before TM1.
- ‘signaling transduction’ – ‘signal transduction’ or just ‘signaling’. Other minor corrections needed for: ‘Although absence of direct evidence’ or ‘A long-standing question of the GPCR field’
- Please, use ‘sequence motif’ instead of ‘barcode’, line 69. And: ‘a plethora of tertiary and quaternary structure information of receptor/G-protein complex determines receptor/G protein type recognition’. If Authors also included a quaternary structure in this sentence, it would be good to provide a reference for this (class B or C?). Could this fragment also include the current state-of-art for G proteins, like in line 255?
- Line 75-77 – emphasize this sentence as it is crucial for the whole manuscript.
- Line 77 – ‘is a cornerstone’
- Line 171 – should be ‘molecular dynamics’
- Line 98 – specify what factor could it be.

- Lines 103-121: although undoubtedly successful, used technology significantly alters sequences of G proteins. Could Authors discuss the impact (or lack of any impact) of these changes on the receptor-G protein interactions? Could these sequence modifications have any influence on better stability of Gq complexes comparing other G types (line 142)?
- Extended Data Fig. 6 – use ‘Replica 1, 2, etc.’ (or ‘Simulation 1, 2, etc.’) instead of ‘MD1, 2, etc.’
- Line 175 – is pi-stacking (F569) also involved?
- Line 176-183 – could it be just because the GPR110 agonist is tethered and thus the receptor itself influences its different orientation comparing gherlin, opioid etc. receptor complexes? Yet, the TM6 bending similar to one in GLP-1R contradicts it a bit.
- Fig. 3c – depict helices and include other residues in this region.
- Line 228 – provide references.
- Line 252-254 – distances rather do not correlate with interface size, why?
- Line 266-267 – does this classification follow any known data for other receptors, e.g., receptors that are only Gq/Gs coupled or only Gi/G12/G13 coupled (add references)?
- Line 456 – is there any justification for such composition of the lipid bilayer?
- Line 466 – correct: ‘pressure (1 atom)’.

REVIEWER COMMENTS

Reviewer #1 (Remarks to the Author):

In this study, Zhu and colleagues determined five structures of adhesion receptor GPR110 in complex with different types of G proteins by cryo-EM. Despite several adhesion receptors have been published on Nature (doi: 10.1038/s41586-022-04619-y, doi: 10.1038/s41586-022-04590-8), this study firstly reported a series of G protein bound activated GPR110 and compared the differences of molecular mechanism of G protein coupling. Structural comparison reveals activation mechanism of GPR110, as well as two classes of G protein subunits in GPR110 complexes. This study is important for structure and pharmacology of GPCR field, but some concerns should be addressed in the future revised version.

Thank you very much for your positive comments on our study.

1. GPR110 exerts its multi-pharmacological functions through different G protein subunits, can the author provide more evidence to elucidate primary signaling pathway of GPR110? Can the author carry out more mutations in ligand binding pocket and test their contribution for different G proteins signaling? If so, could the author design new peptide for biased signaling pathway of GPR110? This is one of the excellent points of this study, I appreciate if the author can offer additional information.

Regarding about more evidence of GPR110 signaling, we have adopted a newly developed BRET assay which can quantitatively measure constitutive activity of GPCR (Science Signaling, 2021, doi/10.1126/scisignal.abf1653). Unlike the reporter assay, the BRET assay directly measure the association between receptor and G-protein, a decrease of BRET signal indicates the separation of $G\alpha$ subunit from $G\beta\gamma$ subunit, a landmark of receptor activation. The data shows that GPR110 is able to couple all 4 four major G-protein pathways, interestingly, the BRET assay data also shows that G_i has the weakest coupling activity to GPR110, while G_q , G_{13} and G_s have a strong coupling activity to GPR110 (Fig. 1 of rebuttal letter and Fig. 6a of revised manuscript), consistent with our structural observation that G_q , G_{13} and G_s complexes have a stronger receptor/G-protein association and a higher quality density map, while the G_i complex has the poorest receptor/G-protein association and density map.

Fig. 1. A BRET assay to examine the constitutive activity of the CTF of GPR110.

To further elucidate the physiological importance of our study on GPR110 signaling, we ask whether the stalk peptide engagement of receptor direct GPR110 signaling. Previous GPR110 studies showed that synaptamide, a metabolic product of DHA, is the ligand of GPR110. However, the authors claimed that the activity of GPR110 is not elicited by the stalk peptide/receptor engagement, instead by GAIN domain/receptor interaction. To test this hypothesis, we added the stalk peptide of GPR110 to the primary mouse neuron, we found that adding of the stalk peptide promotes neurite outgrow, similar to the DHA control (Fig. 2 of rebuttal letter and Fig. 6b-c of revised manuscript). This data is a direct evidence that GPR110 acts through the tethered-peptide mechanism for receptor activation.

Fig. 2. Stalk peptide of GPR110 induces neurite outgrowth. The growth of neuron is measured by the length of axon.

We did test more mutations on the ligand binding pocket of receptor, however, we did not identified mutation that selectively activates one signaling over others. Since the ligand binding pocket is big, a thorough screen of the whole ligand binding pocket will take a very long time, given the importance of our discovery, we think it is our priority to get our study to the research community first.

Since the above primary mutations study is not very successful, we did not design peptide for biased-agonism. However, based on the importance of the stalk peptide on neurite outgrowth. We have plan of developing small molecules targeting the ligand binding pocket of GPR110. However, this will be a separated story and is out of the scope of this paper.

2. The author should generate more mutations to validate the different G protein coupling mode.

We did try to generate more mutations that can selectively activate one signaling over others. Due to the large interface between receptor/G-protein, we currently did not identify a mutation that can selectively activate one signaling over others. However, in a similar study of ADGRL3 (submitted to somewhere else), we did identify mutations that selectively activate one signaling over others. A thorough screen of the interface of GPR110 will take a very long time, we think the first priority is to get our discovery to the community, thus others can joint to this effort.

3. Could the author test the expression level of mutations of GPR110?

We did test the expression of all mutations of our study. The data shows that all mutations expressed at a similar level (Fig. 3 of rebuttal letter and Supplementary Fig. 6f).

Fig. 3. Expression level of GPR110 mutants. All GPR110 constructs were fused to a FLAG tag at their C-terminus. The expression level is measure by anti-FLAG antibody.

Reviewer #2 (Remarks to the Author):

This paper reports the first set of structures of an adhesion GPCR, GPR110, activated by its tether stalk motif and its coupling with five different subtypes of G proteins, Gq, Gs, Gi, G12 and G13. The structures are of high quality with appropriate resolutions, which reveal clear feature of the stalk motif and G protein coupling interactions. The structure information is well supported by mutational studies related to the stalk motif binding pocket and the G protein

couple interface. The manuscript is very well written and the figures are nicely prepared. A similar structure has been published online by Nature recently but this current paper reports the work well beyond what has been reported in Nature. I would recommend its publication ASAP to catch what have been published in Nature this Month, after the minor points below have been addressed.

Thank you very much for your highly positive comments on our study.

1. The title should include adhesion GPCR and stalk motif.

We have changed the title to “Structural basis of adhesion GPCR GPR110 activation by stalk peptide and G_q , G_s , G_i , G_{12} and G_{13} coupling”.

2. The common name of adhesion GPCR for GPR110 should also be in the abstract.

Thank you. We added ADGRF1 in the abstract.

3. The original conserved HTEX motif in class B GPCRs was reported by Yin et al ([https://www.jbc.org/article/S0021-9258\(20\)40795-1/fulltext](https://www.jbc.org/article/S0021-9258(20)40795-1/fulltext)), and the correct citation should be used.

We corrected this error.

4. In reference 40, the authors used the Dopamine receptor D2R as the initial model for their model building. However, this model has poor resolution and density in the TM1 region of the receptor. The correct model of D2R should be 7JVR, which has better resolution with clear density for TM1. It will be good to compare the two D2R structure with GPR110.

Thank you for pointing out this. We did a comparison of 6vms, 7jvr and our GPR110, yes, the TM1 position of 7jvr is closer to that of GPR110 (Fig. 4 and Supplementary Fig. 6c). Because the density map for GPR110 is of high quality, we were able to de novo build the whole TM1 even though there is a big separation between the model TM1 and the final GPR110 TM1.

Fig. 4. A comparison of GPR110 with the initial model of D2R.

Reviewer #3 (Remarks to the Author):

I have no major concerns about the manuscript. Indeed, data for G12/13 in PDB is highly limited comparing Gs or Gi. This confirms that the manuscript includes outstanding findings and the suggestion stated in lines 129-130 even goes far beyond when joined with line 139.

Some minor points are listed below:

- "self-tethering" activation – instead: 'self-activation by tethered agonist' or 'self-cleavage to release tethered agonist' or 'tethered self-activation' would be better.

Thank you for pointing out this! We have changed the "self-tethering" to "self-activation by tethered agonist".

- 'The autoproteolysis also results a split of the receptor into two fragments, the N-terminal fragment (NTF) and C-terminal fragment (CTF). The CTF contains the stalk peptide,' instead: 'The autoproteolysis allows to release of the N-terminal fragment (NTF) of the receptor. The remaining N-terminal region of the receptor includes the stalk peptide.'. Using CTF in this case may unnecessarily suggest that the tethered agonist is formed from the C-terminal region of the receptor and not from its N-terminal region, located before TM1.

Thank you very much for your suggestion. We have corrected this.

- 'signaling transduction' – 'signal transduction' or just 'signaling'. Other minor corrections needed for: 'Although absence of direct evidence' or 'A long-standing question of the GPCR field'

We have corrected those errors. Thank you.

- Please, use 'sequence motif' instead of 'barcode', line 69. And: 'a plethora of tertiary and quaternary structure information of receptor/G-protein complex determines receptor/G protein type recognition'. If Authors also included a quaternary structure in this sentence, it would be good to provide a reference for this (class B or C?). Could this fragment also include the current state-of-art for G proteins, like in line 255?

Thank you for your suggestion. We have replaced the barcode with sequence motif and delete the barcode must be based on a plethora of tertiary and quaternary structure information... to avoid confusion. Our understanding of "code" or sequence motif is beyond the primary sequence, it should contain a tertiary information of the recognition of the last 7 key residues of α H5.

- Line 75-77 – emphasize this sentence as it is crucial for the whole manuscript.

We modified the sentence to reflect the importance of our study. Thank you.

- Line 77 – 'is a cornerstone'

Corrected.

- Line 171 – should be 'molecular dynamics'

Corrected.

- Line 98 – specify what factor could it be.

We add endogenous ligand.

- Lines 103-121: although undoubtedly successful, used technology significantly alters sequences of G proteins. Could Authors discuss the impact (or lack of any impact) of these changes on the receptor-G protein interactions? Could these sequence modifications have any influence on better stability of Gq complexes comparing other G types (line 142)?

Thank you for this question. The idea of using NanoBiT is to increase local concentration of G-protein to receptor, the loop between receptor and the tethered G-protein is long and flexible enough to not interfere the receptor/G-protein interaction (specifically, the intracellular cavity of receptor/ α H5 interaction). More importantly, structures solved by the NanoBiT technology largely agree with structures solved by conventional way. For instance, recently resolved SST2R 7WIC (with NanoBiT) vs 7T10 (without NanoBiT), S1PR1/BAF312/ G_i complex by our group 7EO4 (with NanoBiT) vs that by other group 7EVY (without NanoBiT). Particularly, in this case, a comparison of the GPR110/ G_s , G_i complexes resolved by our group (with NanoBiT) vs the reported ADGRF1/ G_s , G_i complexes (without NanoBiT) (Supplementary Fig. 8), shows that they are almost identical. Also, the NanoBiT strategy works on all G-proteins, not just G_q protein, the higher resolution of the GPR110/ G_q complex is due to the nature of receptor/ G_q interaction.

- Extended Data Fig. 6 – use ‘Replica 1, 2, etc.’ (or ‘Simulation 1, 2, etc.’) instead of ‘MD1, 2, etc.’

Corrected, thank you.

- Line 175 – is pi-stacking (F569) also involved?

Yes, it forms a π - π interaction with F641. We added a description of this in the main text.

- Line 176-183 – could it be just because the GPR110 agonist is tethered and thus the receptor itself influences its different orientation comparing gherlin, opioid etc. receptor complexes? Yet, the TM6 bending similar to one in GLP-1R contradicts it a bit.

We do not think that the unique orientation of GPR110 is solely caused by the tethered peptide as GPR52 and PAR1-2 also has a tethered peptide mechanism.

- Fig. 3c – depict helices and include other residues in this region.

Added, thank you.

- Line 228 – provide references.

Added.

- Line 252-254 – distances rather do not correlate with interface size, why?

The overall interface size include areas are weakly associated. The strength of interaction is mainly determined by the tight association. Although the α H5 of G13 is tightly associated with

TM3/TM4 (closest distance), the overall interface includes other area (i.e. αN) that are loosely associated with receptor, so its interface is not the biggest.

- Line 266-267 – does this classification follow any known data for other receptors, e.g., receptors that are only Gq/Gs coupled or only Gi/G12/G13 coupled (add references)?

We are the first one who compare 5 major couplings on the same receptor, the classification is solely based on the structural observations of the locations and interactions of the last 8 residues of $\alpha H5$ in the engagement, therefore there is no literature available.

- Line 456 – is there any justification for such composition of the lipid bilayer?

This is a commonly used lipids composition (POPC:CHR = 4:1) for GPCR and some channel in MD simulation.

- Line 466 – correct: 'pressure (1 atm)'.

corrected, 1atm.

REVIEWERS' COMMENTS

Reviewer #1 (Remarks to the Author):

The revised manuscript has addressed my concerns. I would like to suggest to publish the manuscript as soon as possible.